# Inhibition of Mitochondrial Dynamics Preferentially Targets Pancreatic Cancer Cells with Enhanced Tumorigenic and Invasive Potential

**DOI:** 10.3390/cancers13040698

**Published:** 2021-02-09

**Authors:** Sarah Courtois, Beatriz de Luxán-Delgado, Laure Penin-Peyta, Alba Royo-García, Beatriz Parejo-Alonso, Petra Jagust, Sonia Alcalá, Juan A. Rubiolo, Laura Sánchez, Bruno Sainz, Christopher Heeschen, Patricia Sancho

**Affiliations:** 1Translational Research Unit, Hospital Universitario Miguel Servet, IIS Aragon, 50009 Zaragoza, Spain; scourtois@iisaragon.es (S.C.); albaroyo97@gmail.com (A.R.-G.); bparejo@iisaragon.es (B.P.-A.); 2Centre for Stem Cells in Cancer & Ageing, Barts Cancer Institute, Queen Mary University of London, London EC1M 6BQ, UK; b.deluxandelgado@qmul.ac.uk (B.d.L.-D.); laurepeyta@gmail.com (L.P.-P.); petra.jagust@gmail.com (P.J.); 3Department of Biochemistry, School of Medicine, Instituto de Investigaciones Biomédicas (IIBm) “Alberto Sols” CSIC-UAM, Autónoma University of Madrid (UAM), 28029 Madrid, Spain; sonia.alcala@uam.es (S.A.); bruno.sainz@uam.es (B.S.J.); 4Chronic Diseases and Cancer, Area 3, Instituto Ramón y Cajal de Investigación Sanitaria (IRYCIS), 28029 Madrid, Spain; 5Department of Zoology, Genetics and Physical Anthropology, Veterinary Faculty, Universidad de Santiago de Compostela, 27002 Lugo, Spain; ja.rubiolo@usc.es (J.A.R.); lauraelena.sanchez@usc.es (L.S.); 6Center for Single-Cell Omics, State Key Laboratory of Oncogenes and Related Genes, Shanghai Jiao Tong University School of Medicine, Shanghai 200025, China

**Keywords:** mitochondria, mitochondrial dynamics, mitochondrial fission, cancer stem cells, CD133, DRP1, energy crisis, PDAC, pancreatic cancer

## Abstract

**Simple Summary:**

Due to their intrinsic aggressiveness, cancer stem cells (CSCs) represent an essential target for the design of effective treatments against pancreatic cancer, one of the deadliest tumors. As pancreatic CSCs are particularly dependent on the activity of their mitochondria, we here focus on mitochondrial dynamics as a critical process in the homeostasis of these organelles. We find that pancreatic CSCs rely on mitochondrial fission, and its pharmacological inhibition by mDivi-1 resulted in the accumulation of dysfunctional mitochondria, provoking an energy crisis and cell death in this subpopulation. Consequently, mDivi-1 blocked cellular functions related to cancer aggressiveness such as in vivo tumorigenicity, invasiveness, and chemoresistance. Our data suggest that the inhibition of mitochondrial fission represents a promising target for designing new multimodal therapies to fight pancreatic cancer.

**Abstract:**

Pancreatic ductal adenocarcinoma (PDAC) is one of the deadliest tumors, partly due to its intrinsic aggressiveness, metastatic potential, and chemoresistance of the contained cancer stem cells (CSCs). Pancreatic CSCs strongly rely on mitochondrial metabolism to maintain their stemness, therefore representing a putative target for their elimination. Since mitochondrial homeostasis depends on the tightly controlled balance between fusion and fission processes, namely mitochondrial dynamics, we aim to study this mechanism in the context of stemness. In human PDAC tissues, the mitochondrial fission gene *DNM1L* (DRP1) was overexpressed and positively correlated with the stemness signature. Moreover, we observe that primary human CSCs display smaller mitochondria and a higher DRP1/MFN2 expression ratio, indicating the activation of the mitochondrial fission. Interestingly, treatment with the DRP1 inhibitor mDivi-1 induced dose-dependent apoptosis, especially in CD133^+^ CSCs, due to the accumulation of dysfunctional mitochondria and the subsequent energy crisis in this subpopulation. Mechanistically, mDivi-1 inhibited stemness-related features, such as self-renewal, tumorigenicity, and invasiveness and chemosensitized the cells to the cytotoxic effects of Gemcitabine. In summary, mitochondrial fission is an essential process for pancreatic CSCs and represents an attractive target for designing novel multimodal treatments that will more efficiently eliminate cells with high tumorigenic potential.

## 1. Introduction

Pancreatic ductal adenocarcinoma (PDAC), the most prevalent form of pancreatic cancer, is the third most frequent cause of cancer-related deaths nowadays [1]. Considering its rising incidence, extreme aggressiveness, and the lack of effective treatments [2], pancreatic cancer is predicted to become the second most frequent cause of deaths caused by cancer by 2030 [3].

The main malignant features of PDAC, such as chemoresistance to conventional systemic therapies, rapid relapse after treatment and metastasis formation in vital organs, such as the liver and lungs, can be attributed to specific subpopulations of cancer cells with tumor- and metastasis-initiating properties, known as pancreatic cancer stem cells (CSCs) [4,5,6,7]. Although they represent a small fraction of the cancer cell population within the tumor, CSCs are the main drivers of tumorigenesis in the pancreas and metastatic sites, due to their self-renewal capacity and differentiation into rapidly proliferating cancer cells. Additionally, their combined chemoresistance and tumorigenic capacity make them responsible for disease relapse [4,6,8]. Therefore, identification of CSCs vulnerabilities is essential in order to design more effective therapies against PDAC.

We recently discovered that pancreatic CSCs are particularly sensitive to mitochondrial targeting, due to their extreme dependence on oxidative phosphorylation (OXPHOS) [9]. Essentially, pancreatic CSCs tightly control the expression balance between the glycolysis-promoting oncogene c-MYC and the mitochondrial biogenesis transcription factor PGC1-α, favoring mitochondrial metabolism in order to maintain full stemness. In fact, we have demonstrated that perturbing mitochondrial function by either inhibition of the electron transport chain (ETC) with the antidiabetic agent metformin [9,10] or altering its redox state [9,11] significantly decreased pancreatic CSCs functionality and chemoresistance. Thus, our results identified mitochondrial activity as a key vulnerability for pancreatic CSCs.

We have recently demonstrated that mitochondrial biogenesis driven by PGC1-α, and recycling of dysfunctional mitochondria through ISGylation-mediated mitophagy are essential processes for pancreatic CSCs [9,12]. In between these initial and final steps in the lifecycle of mitochondria, fusion and fission represent the main events involved in mitochondrial dynamics. Fusion and fission processes are controlled by different sets of members of the Dynamin family, in conjunction with several adapter proteins. Specifically, dynamin-related/-like protein 1 (DRP1), dynamin 2 (DNM2), mitochondrial fission 1 (FIS1), and mitochondrial fission factor (MFF) are involved in the fission process, where one mitochondrion divides into two daughter mitochondria. On the other hand, mitofusins 1 and 2 (MFN1 and MFN2) and optic atrophy 1 (OPA1) control mitochondrial fusion, where two mitochondria form one mitochondrion. The balance between these dynamic transitions regulates size, number, distribution, and quality control of mitochondria, and therefore are key to maintain their correct functionality [13]. Notably, mitochondrial dynamics are essential for successful asymmetrical division in normal stem cells (SC) [14] and have been linked to proliferation and survival of stem cells in normal tissues and some cancer types [15,16].

Although increased mitochondrial fission has recently been linked to metabolic changes induced by mutant KRAS in PDAC [17,18], our knowledge is still sparse concerning the relationship between mitochondrial dynamics and stemness in this cancer type. Here, we show that mitochondrial fission is particularly relevant for pancreatic CSCs, and its inhibition with the compound mDivi-1 effectively diminishes CSC content in PDAC patient-derived xenografts (PDXs). Indeed, mDivi-1 treatment resulted in the accumulation of dysfunctional mitochondria that provoked energy crisis and apoptosis in CSCs. Finally, mDivi-1 inhibited stemness-related properties such as self-renewal, tumorigenicity, and invasiveness and enhanced the toxicity of Gemcitabine, suggesting that inhibition of mitochondrial fission may represent an attractive target for the design of novel combinatory therapeutic strategies for PDAC treatment.

## 2. Materials and Methods

### 2.1. Cell Culture and Patient-Derived Xenografts

PDAC patient-derived xenografts (PDAC PDX: 185, 215, 253 and 354) were obtained through the Biobank of the Spanish National Cancer Research Centre (CNIO), Madrid, Spain (references CNIO20-027, I409181220BSMH, 1204090835CHMH). Dissociation and establishment of in vitro cultures were performed as previously described [19], and cells were maintained for a maximum of 15 passages. PDXs were grown in RPMI (61870044) supplemented with 10% FBS and 50 U/mL penicillin/streptomycin (all from Gibco, Life Technologies, Carlsbad, CA, USA). For actual experiments, medium was switched to DMEM/F12 (31331028) supplemented with 2% B27, 50 U/mL penicillin/streptomycin (all from Gibco) and 20 ng/mL bFGF (Pan-Biotech, GmbH, Aidenbach, Germany). HPDE (Human Pancreatic Duct Epithelial Cell Line) cells were grown in Keratinocyte Serum Free Media supplemented with bovine pituitary extract and EGF (Gibco). HFF (Human foreskin fibroblasts) were grown in DMEM supplemented with 10% FBS (61965026, Gibco). All the cells were grown at 37 °C in a humidified 5% CO_2_ atmosphere.

### 2.2. Treatments

mDivi-1 was dissolved in DMSO following the manufacturer’s instructions (S7162, Selleckchem, Munich, Germany). Depending on the experimental design, cells were treated for 24 h to 7 days, at concentrations ranging from 10 to 80µM. DMSO compensation was included in all the conditions to reach the DMSO concentration added in the higher concentration used for each experiment (0.8% DMSO when the maximal concentration used was 40 µM, 1.6% DMSO when the maximal concentration used was 80 µM). Gemcitabine diluted in 0.9% sodium chloride (Eli Lilly, Indianapolis, IN, USA) was used as standard chemotherapy with concentrations ranging from 1 nM to 5 µM. For experiments in hypoxia, cells were maintained in 3% O_2_ in a ICO50med incubator (Memmert GmbH, Schwabach, Germany).

### 2.3. Human Data Analysis

Expression data from human PDAC tissue and normal tissue were analyzed using the webserver GEPIA2 (TCGA and the GTEx project databases; http://gepia2.cancer-pku.cn/) accessed on 2 December 2019 [20] or OncomineTM (Badea, Buchholz, Grutzmann, Iacobuzio-Donahue, Ishikawa, Logsdon, Pei, Segara databases). The Pearson correlation coefficient was calculated to study the association of *DNM1L* gene with a stemness signature defined by the combined expression of the pluripotency-related genes *NANOG*, *KLF4*, *SOX2* and *OCT4*. For disease-free survival analysis, the Hazard Ratio (HR) was calculated in GEPIA2 using the Cox Proportional Hazards model for pancreatic cancer patients from the respective upper and lower quartiles of expression of the indicated genes.

### 2.4. Transmission Electron Microscopy (TEM)

PDAC 185 or 354 cells were used for TEM experiments. As described in [21], 185 cells were sorted for CD133 expression and pellets were fixed with 0.1 M cacodylate buffer with a pH of 7.4 at room temperature. Sections were processed by the USC Electron Microscopy unit (Lugo, Galicia) per standard protocols. Pictures were taken with a JEM-1010 transmission electron microscope (JEOL, Tokyo, Japan and analyzed by Adobe PhotoShop CS4 EXTENDED V11.0 (Adobe Systems, Montain View, CA, USA). 354 cells were seeded at 40,000 cells per compartment in 8-wells microscopy slides in 200µL of medium and treated with mDivi-1 at 40 µM. After 48 h cells were fixed in glutaraldehyde (16210, Electron Microscopy Science, Hatfield, UK) and samples processed by the Electron Microscopy of Biological Systems Unit at the University of Zaragoza following standard procedures. Samples were visualized on a JEOL JEM 1010 100 kV microscope (JEOL, Tokyo, Japan). Mitochondrial area quantification was performed using ImageJ.

### 2.5. Immunoblots

Treated cells were lysed in RIPA buffer (R0278, Sigma-Aldrich, St. Louis, MO, USA) supplemented with protease inhibitors (J64156) and phosphatase inhibitors (J61022) (Alfa Aesar, Thermo Fisher Scientific, Waltham, MA, USA). Proteins were quantified using the Pierce™ BCA Protein Assay Kit (23225, Thermo Fisher Scientific). After the electrophoresis process in 10% Tris-Glycine gels (XP001002, Invitrogen), proteins were transferred to a PVDF membrane (88518, Thermo Fisher Scientific) and incubated overnight at +4 °C with the different primary antibodies, listed below. After washes with PBS-Tween 0.1%, the membranes were incubated with peroxidase-conjugated goat anti-rabbit or goat anti-mouse secondary antibodies (656120 and A10685 respectively, Invitrogen). Bound antibody complexes were detected using Pierce™ ECL Western Blotting Substrate (32109, Thermo Fisher Scientific) and visualized on CL-X Posure^TM^ Films (34091, Thermo Scientific). Bands intensities were analyzed with ImageJ software and normalized to β-actin.

Antibodies against DRP1 (8570S, dilution 1:1000), MFN2 (11925S, dilution 1:1000), LC3B (3868S, dilution 1:3000), P-AMPKα (2531S, dilution 1:1000), AMPKα (2532S, dilution 1:1000), were obtained from Cell Signaling Technology (Denvers, MA, USA). The OXPHOS Human WB Antibody Cocktail (M5601-360, dilution 1:2000) was obtained from Abcam (Cambridge, UK), and β-actin, clone AC-74 from Sigma Aldrich (A2228, dilution 1:10,000).

### 2.6. RNA Extraction and Quantitative Reverse Transcription Polymerase Chain Reaction (RTqPCR)

Total RNA was extracted using TRIzol kit (Life Technologies) according to the manufacturer’s instructions. Retrotranscription was performed with 1µg of total RNA using SuperScript II reverse transcriptase (Life Technologies) and random hexamers. Quantitative polymerase chain reaction (qPCR) was performed using PowerUp SYBR Green master mix (Applied biosystems, Thermo Fisher Scientific), according to the manufacturer’s instructions. The primers used are detailed in Table 1. *HPRT* was used as endogenous housekeeping control.

### 2.7. Proliferation Assay

In total, 10,000 cells were seeded in triplicates in different 96-well plates and treated 24 h later in 200 µL of supplemented DMEM/F12 containing different concentrations of mDivi-1 and/or Gemcitabine. After 3 and 7 days of treatment, cells were stained with 2% crystal violet (CV) (405831000, Acros Organics, Fisher Scientific) and dried. The CV absorbance was assessed after dissolution in SDS (1% in PBS) at 590 nm. The proliferation rate was normalized to control conditions, set to 100%.

### 2.8. Cytotoxicity Assay

Cytotoxicity assays were performed using the MultiTox-Fluor Multiplex Cytotoxicity Kit (G9201, Promega, Madison, WI, USA) following the manufacturer’s instructions.

### 2.9. Flow Cytometry Analysis and Sorting

After treatment, cells were trypsinized, washed once in PBS, and resuspended in Blocking Buffer (2% FBS, 0.5% BSA in PBS) for 15 min on ice under agitation. Cells were stained for 30 min at +4 °C with APC or PE-conjugated anti-CD133 antibodies (diluted at 1/200 or 1/400 respectively; Biolegend, San Diego, USA) or corresponding control Immunoglobulin G1 (IgG1, Biolegend) antibody as a control for non-specific staining. After washes, pellets were resuspended in PBS with MitoTracker^TM^ Deep Red FM (MT) (m22426, Life Technologies), MitoStatus TMRE (564696, BD Biosciences, San Jose, CA, USA) or Cell ROX Deep Red reagent (1691766, Life Technologies) for 20 min at room temperature. Annexin V-APC staining was performed on attached and floating cells according to manufacturer’s instructions (550474 & 556454, BD Biosciences, San Diego, CA, USA). Zombie Violet Dye (77477, Biolegend, San Diego, CA, USA) was used to exclude non-viable cells. 50,000 cells per sample were analyzed using a FACS Canto II (BD, Franklin Lakes, NJ, USA) and analyzed with FlowJo 9.2 software (Ashland, OR, USA). Moreover, viable cells corresponding to the CD133 negative and positive populations were sorted into 5 mL tubes containing full RPMI medium using a SONY SH800S instrument (SONY, Tokyo, Japan).

### 2.10. XF Extracellular Flux Analyzer Experiments

In total, 30,000 cells per well were plated in XF96 Cell Culture Microplates (Seahorse Bioscience, Agilent, Santa Clara, CA, USA) previously coated with Cell-Tak (BD Biosciences). For OCR determination, cells were incubated for 1 h in base assay medium (D5030, Sigma Aldrich, Merck, Darmstadt, Germany) supplemented with 2 mM glutamine, 10 mM glucose, and 1 mM pyruvate, prior to OCR measurements using the XF Cell Mito Stress Test Kit (Seahorse Bioscience). Concentrations for Oligomycin and FCCP were adjusted for each primary cell type as follows: Oligomycin, 1.2 mM for 215 and 253; and 0.8 μM for 354 cells; FCCP 1.2 μM for 215 and 253; and 0.4 μM for 354 cells. Oligomycin, FCCP, Rotenone and Antimycin A were dissolved in DMSO. For glycolytic metabolism measurements, cells were incubated in basal media prior to injections using the XF Glyco Stress Test kit (Seahorse Bioscience). For the evaluation of the acute response to mDivi-1, different concentrations of mDivi-1 were injected in ports A–C, and the percentage of complex I inhibition was calculated as the percentage of OCR inhibited upon mDivi-1 injection with respect to the inhibition obtained with Rotenone, the latter used as 100%, as described previously [9]. Experiments were run in a XF96 analyzer (Seahorse Bioscience), and raw data were normalized to protein content using the Pierce™ BCA Protein Assay Kit (Thermo Fisher Scientific).

### 2.11. ATP Measurement

After treatment, cell pellets were washed with PBS and then resuspended in ultra-pure water (10977035, Invitrogen). ATP was quantified using the ATP Determination Kit (A22066, Invitrogen) following the manufacturer’s instructions. Normalization of data was performed using protein concentrations measured on the same samples with the Pierce™ BCA Protein Assay Kit.

### 2.12. Sphere Formation Assay

Ten thousand cells were seeded, with or without treatment, in non-adherent plates (ultra-low attachment plates (3473, Corning, NY, USA) or normal plates previously coated with a 10% poly-2-hydroxyethylmathacrylate (polyHEMA, Sigma)), in 1 mL of supplemented DMEM/F12. After 7 days, formed spheres were counted using an inverted microscope with 20× magnification.

### 2.13. Colony Formation Assay

Either 500 or 1000 cells per well were seeded in 2 mL of supplemented DMEM/F12 with treatments. Media and treatments were refreshed every 7 days. After 21 days, cells were stained with crystal violet and the number of colonies was manually counted.

### 2.14. Invasion Assay

Invasion assays were performed using 354 cells pretreated for 48 h with mDivi-1 40µM in supplemented DMEM/F12 or conditioned media from M2-polarized macrophages (MCM), previously demonstrated to induce EMT and stemness in our PDX models [22]. MCM was obtained as follows: leucocyte cones from anonymous healthy donors were obtained from the National Blood Transfusion Service (UK) as approved by the City and East London Research Ethics Committee (17/EE/0182). Cones were stored at 4 °C and used within 24 h of delivery to maintain cell viability. Monocyte-derived human macrophage culture, polarization into M2-like macrophages and generation of conditioned medium were performed as previously described [22]. In brief, monocyte-derived human macrophage cultures were maintained in IMDM (Gibco), supplemented with 10% human AB serum, and polarized by incubation with 0.5 ng/mL of macrophage colony-stimulating factor for 48 h (MCSF; PeproTech, Rocky Hill, NJ, USA). To obtain conditioned media, macrophages were washed with PBS and cultured for additional 48 h in supplemented DMEM:F12 (see Section 2.1). Media was then collected, centrifuged, and the supernatant stored at −80 °C.

After treatments, cells were trypsinized and counted, and 150,000 cells were seeded on top of 8.0 µm PET membrane invasion chambers coated with growth factor reduced Matrigel (354480, Corning, NY, USA) in serum free media. After 24 h, the invasion of cells towards 20% FBS was assessed after fixation with 4% formaldehyde and staining with crystal violet. Five fields per well were manually counted at 20× magnification.

### 2.15. Wound Healing Assay

For the wound healing assay, 50,000 cells per well were seeded in ImageLock 96 w plates (IncuCyte^®^ technology). Once the cells were confluent, scratches were performed and, after washing with PBS, cells were incubated in supplemented DMEM/F12 or MCM media with or without different concentrations of mDivi-1. Cell migration was monitored using IncuCyte^®^ Live-Cell Analysis System (Sartorius, Göttingen, Germany). The percentage of wound closure was calculated using the IncuCyte^®^ Software.

### 2.16. In vivo Extreme Limiting Dilution Assay (ELDA)

In total, 354 cells were treated with 40 µM of mDivi-1 for 72 h, trypsinized and resuspended in supplemented DMEM/F12 with Matrigel (50:50). Two doses of cells (1000 or 10,000 cells), were subcutaneously injected in both flanks of 6 weeks-old nude (Foxn1nu) male and female mice (*n* = 6 mice per group). Tumor size was measured using a caliper and followed for 10 weeks, when the control tumors had reached the humane endpoint. The ELDA calculation was performed at http://bioinf.wehi.edu.au/software/elda/ accessed on 28 January 2020. Mice were housed according to institutional guidelines and all experimental procedures were performed in compliance with the institutional guidelines for the welfare of experimental animals as approved by the Universidad of Zaragoza Ethics Committee (CEICA PI22/17) and in accordance with the guidelines for Ethical Conduct in the Care and Use of Animals as stated in The International Guiding Principles for Biomedical Research involving Animals, developed by the Council for International Organizations of Medical Sciences (CIOMS).

### 2.17. Statistical Analysis

Data are presented as the mean ± S.E.M. A Mann–Whitney test or Student’s *t*-test were used for two group comparisons and a Kruskal–Wallis test or one-way analysis of variance (ANOVA) for multiple comparisons. Data were analyzed using GraphPad Prism and differences were considered significant at *p* < 0.05.

## 3. Results

### 3.1. Mitochondrial Fission Is Associated with Stemness and Epithelial-to-Mesenchymal Transition in Human PDAC

Firstly, a bioinformatic analysis using the webserver GEPIA2 was performed to determine the relative expression of the main genes regulating mitochondrial dynamics in transcriptional data from normal human pancreas and PDAC tissues included in the TCGA and GTEx projects (*DNM1L*, *DNM2*, *FIS1*, and *MFF* for fission; *MFN1*, *MFN2*, and *OPA1* for fusion) (Figure 1A). Except for *MFF* and *MFN1*, all other genes involved in mitochondrial dynamics showed a statistically significant upregulation in PDAC tumoral tissue compared to normal tissue. Differences remained significant when the genes were combined together as a mitochondrial dynamics signature. Interestingly, PDAC patients expressing this signature had a lower survival (Appendix A).

Next, we interrogated eight additional PDAC datasets to study the expression of these genes in tumor vs. normal tissue. Although the results varied across datasets, *DNM1L* and *MFF*, both related to mitochondrial fission, were consistently and significantly overexpressed in PDAC tissue (Figure 1B). These results support the implication of the mitochondrial fission process in PDAC and are in line with recent studies demonstrating a crucial role of mitochondrial fission in metabolic changes related to mutant KRAS in PDAC [17,18]. Notably, *DNM1L* expression strongly correlated with gene signatures associated with aggressiveness in PDAC: the stemness signature routinely used by our group (*NANOG*, *OCT4*, *KLF4*, *SOX2*; Appendix A) and the epithelial-to-mesenchymal transition (EMT) signature formed by *ZEB1*, *SNAI1*, and *SNAI2* (Appendix A). Indeed, the joint overexpression of *DNM1L* and the above stemness signature predicted a decreased overall survival in PDAC patients (Appendix A). Overall, these results suggested an association of mitochondrial fission and signatures related to pancreatic CSCs and aggressiveness.

We next aimed to further validate the putative relationship between mitochondrial dynamics and stemness using four different PDX-derived primary cultures. We selected DRP1 (encoded by *DNM1L*) and MFN2 as representative proteins for mitochondrial fission or fusion, respectively. As shown in Figure 1C, the DRP1/MFN2 expression ratio was increased in CSC-enriched conditions (cells grown as spheres or CD133^+^ sorted cells) compared to their differentiated counterparts (cells grown in adherence or CD133^−^ sorted cells), suggesting a strong activation of the mitochondrial fission process in CSCs. As we have previously demonstrated [9], the total mitochondrial mass as determined by flow cytometry was higher in CD133^+^ cells compared to CD133^−^ cells (Figure 1D). However, the combined mitochondrial area as assessed by transmission electron microscopy (TEM) was significantly lower in CD133^+^ cells compared to CD133^−^ cells, in line with increased mitochondrial fission activity in pancreatic CSCs (Figure 1E and Appendix A). Consistent results were observed for cells treated with conditioned media from M2-polarized macrophages (macrophage conditioned media, MCM), previously demonstrated to induce EMT and stemness in our PDX-derived models [22,23] (Appendix A). Together, our results demonstrate that the enhanced mitochondrial fission process in PDAC is mostly confined to the CSC compartment.

### 3.2. The DRP1 Inhibitor mDivi-1 Induces Apoptosis in Primary Pancreatic Cancer Cells, Especially Affecting the CD133^+^ Subpopulation

Considering our results, we aimed to inhibit the mitochondrial fission process with the selective inhibitor of DRP1 mDivi-1 as a novel approach to target pancreatic CSCs, representing cells with enhanced tumorigenic and invasive potential. Using four PDX-derived models, the EC50 for mDivi-1 on PDAC proliferation was >32 µM for three days of treatment and ranged from 18 to 55 µM for seven days of treatment (Appendix A). Expectedly, cells became more resistant to the drug when cultured in hypoxia, as they switch to a non-mitochondrial oxygen-independent metabolism (Appendix A). Therefore, subsequent experiments were performed using mDivi-1 concentrations between 10 and 80 µM.

Next, using the indicated doses of mDivi-1, a series of cytotoxicity assays were performed on two different PDX-derived cultures and two non-tumoral cell lines (i.e., normal pancreatic ductal epithelial cells (HPDE) and human foreskin fibroblasts (HFF)) (Figure 2A). Interestingly, only PDAC cells were sensitive to the drug, even at the highest doses, suggesting that maintenance of a high rate of mitochondrial fission is only critical for PDAC cells. Interestingly, 72 h of treatment with mDivi-1 induced a significant and dose-dependent increase in cell death in PDX-derived cells (Figure 2B), which was particularly evident for CD133^+^ cells (Figure 2C). Indeed, CD133^+^ cells were significantly more sensitive to mDivi-1 than CD133^–^ cells, with 70% cell death for 80 µM of mDivi-1 in CD133^+^ compared to 40% for CD133^–^ cells (Figure 2C), therefore resulting in a marked decrease of the CD133^+^ CSC content (Figure 2D,E). In summary, our results suggest that the inhibition of mitochondrial fission is toxic for PDAC cells, particularly for the CD133^+^ subpopulation.

### 3.3. mDivi-1 Treatment Disrupts Mitochondrial Function

In order to understand why the inhibition of mitochondrial fission was particularly toxic for pancreatic CSCs, we studied the effects of mDivi-1 treatment on mitochondrial metabolism, a well-known vulnerability of these cells, which we have previously described [9]. As an expected direct consequence of blocking mitochondrial fission, mDivi-1 treatment significantly increased the mitochondrial size compared to the control condition (Figure 3A). This effect was accompanied by an increase in mitochondrial mass per cell (Appendix A), which was statistically significant for CD133^+^ cells only (Appendix A). However, we could not detect a significant increase in the expression of mitochondrial respiratory chain complexes as assessed by Western blot (Appendix A). Importantly, many mitochondria exhibited a disorganized internal structure following mDivi-1 treatment (Figure 3B), suggesting a likely accumulation of damaged or unhealthy mitochondria.

Accordingly, mitochondrial activity as assessed by the probe TMRE cells was disturbed, although no clear diminishing pattern could be observed for the bulk cancer cell population (Figure 3C). Still, mDivi-1 treatment was accompanied with doubled Reactive Oxygen Species (ROS) production in bulk cancer cells (Figure 3E). Notably, when studying the effect of mDivi-1 on subpopulations of cancer cells, we found that the mitochondrial membrane potential remained mostly unchanged in CD133^–^ cells, whereas a significant drop could be noted exclusively for CD133^+^ cells (Figure 3D). This diminished mitochondrial functionality in CD133^+^ cells resulted in even more pronounced ROS accumulation (Figure 3F). Since CSCs are particularly sensitive to inhibition of mitochondrial function and oxidative damage [9,11], these treatment effects of mDivi-1 could, at least in part, explain its marked toxicity in CSCs. Notably, mDivi-1 treatment also increased the expression of LC3B (Appendix A), which suggests the activation of autophagy as a counteractive mechanism to avoid the excessive accumulation of defective mitochondria; however, as we have previously shown [12], activation of autophagy does not necessarily ensure mitophagy, especially not when fission is inhibited.

The intricate balance between the mitochondrial biogenesis factor PGC-1α and c-MYC, jointly controlling the metabolic phenotype and stemness of PDAC cells [9], was altered upon mDivi-1 treatment. The balance had shifted towards a more glycolytic phenotype with increased *MYC* expression and subsequently decreased expression of *PGC1A* (Figure 4A), most likely to compensate for the deleterious effects of accumulating defective mitochondria. Therefore, we next studied the effects of mDivi-1 on the oxygen consumption rate (OCR). Indeed, mDivi-1 treatment dose-dependently decreased both basal oxygen consumption and maximal respiration (Figure 4B), as well as ATP-linked OCR at the highest doses (Figure 4C). Importantly, acute injection of mDivi-1 did not result in a consistent inhibition of OCR consumption (Appendix A), arguing against a possible unspecific inhibition of the mitochondrial respiration as previously suggested [24]. Importantly, the observed drop in ATP-linked respiration translated into a significant drop in ATP content for CSC-enriched cultures only (spheres, Figure 4D). Notably, these cells were not capable of consistently increasing glycolysis (differences significant only for 253 cells at 10 and 80 μM, respectively) in order to compensate for the loss of ATP production upon mitochondrial inhibition (Appendix A), even though *MYC* expression had increased (Figure 4A). Consistently, this ATP drop in CSC-enriched cultures led to the activation of the AMP-activated protein kinase (AMPK), the main sensor of cellular energy homeostasis, indicative of energy stress (Figure 4E). Of note, the observed increase in the phospho-AMPK/AMPK ratio was mainly due to the downregulation of total AMPK upon mDivi-1 treatment.

Overall, our results demonstrate that dysregulation of mitochondrial dynamics in response to mDivi-1 treatment led to the accumulation of defective mitochondria in PDAC cells, which translated specifically into an energy crisis in the CSC compartment.

### 3.4. mDivi-1 Treatment Blocks CSC Functionality

Since we demonstrated that treatment with mDivi-1 was particularly toxic for CSCs due to their metabolic peculiarities, we next performed diverse functional assays to analyze the different features associated with stemness and aggressiveness in response to mDivi-1 treatment.

We first checked the expression of stemness-related genes after treatment with mDivi-1, which revealed that *SOX2* downregulation was the only common event when comparing two different PDX models (Figure 5A). Next, we studied the self-renewal capacity of PDAC cells in response to mDivi-1 using sphere and colony formation assays. mDivi-1 treatment dose-dependently reduced the number of formed spheres by >50% compared to the control condition (Figure 5B). Furthermore, mDivi-1 essentially inhibited the ability of treated cells to form colonies at the highest dose tested (Figure 5C).

Finally, to explore the effect of mDivi-1 on tumorigenicity and CSC content, we performed in vivo extreme limiting dilution assays (ELDA) (Figure 5D). Two doses of cells (10^3^ or 10^4^ cells) pre-treated for three days with 40 µM of mDivi-1 were subcutaneously injected into immunocompromised mice and tumor formation was followed for 10 weeks. Interestingly, cells pre-treated with mDivi-1 only formed a single small tumor in each group and, as such, the estimated content of CSCs decreased more than 10-fold (from 1/1425 to 1/20,041 cells; *p* < 0.001) (Figure 5D). Overall, these results demonstrated that the inhibition of mitochondrial fission via mDivi-1 treatment functionally impaired in vitro self-renewal and in vivo tumorigenicity.

We next tested mDivi-1 treatment effects on CSC properties that define the aggressiveness of PDAC, e.g., invasiveness and chemoresistance. PDAC cell migration and invasion was induced by MCM, which were considerably inhibited by mDivi-1 treatment at 40 µM (Figure 6A,B). Interestingly, mDivi-1 did not reverse the MCM-induced upregulation of EMT genes (Figure 6C), suggesting that inhibition of mitochondrial fission does not interfere with the EMT genetic program per se, but rather impedes downstream cellular functions.

Finally, considering the essential role of mitochondria in chemoresistance across different cancer types [25] and the intrinsic chemoresistance of CSCs, we studied the response to Gemcitabine treatment in the context of mitochondrial fission inhibition. First, mDivi-1 treatment synergized with Gemcitabine for inhibiting cell proliferation (Figure 6D), an effect especially noticeable in 215 cells which were totally resistant to Gemcitabine alone. Even more importantly, we observed a summatory effect for the two drugs on the ability of the cells to form spheres (Figure 6E). In general, the addition of mDivi-1 improved the response to Gemcitabine alone, suggesting that this combined treatment strategy could be more efficient in targeting chemoresistant cells.

In summary, our results demonstrate that the inhibition of mitochondrial fission efficiently targets pancreatic CSCs with pronounced aggressive features, such as enhanced tumorigenicity, invasiveness, and chemoresistance, through the accumulation of defective mitochondria and subsequent energy crisis leading to loss of stemness and cell death.

## 4. Discussion

Emerging evidence indicates that mitochondrial fission, and particularly DRP1, are involved in the pluripotency and functionality of SCs [14,15,26]. For instance, genetic modulation of *DNML1* expression or DRP1 activity modified the differentiation state of embryonic SCs or reprogrammed fibroblasts, pushing cells into differentiation or pluripotency, respectively [26,27]. In line with previous reports, our present data now demonstrate a similarly close relationship between DRP1 and stemness for pancreatic cancer. CSCs from glioblastoma were shown to depend on DRP1 activity for growth and self-renewal and, importantly, DRP1 phosphorylation correlated with patient survival [28]. Similarly, we found a positive correlation between *DNM1L* expression and stemness-related signatures in human PDAC samples, resulting in a novel stemness signature that is capable of predicting patients’ outcome (Appendix A). Importantly, we further corroborated these expression data using our PDAC models and demonstrated that pancreatic CSCs accumulate significantly smaller mitochondria, correlating with an increased DRP1/MFN2 ratio, both indicative of increased mitochondrial fission activity (Figure 1C–E).

It is well accepted that elevated fission activity results in mitochondrial fragmentation and impaired OXPHOS, whereas increased fusion activity leads to enhanced oxidative metabolism [29]. However, as opposed to normal SCs featuring glycolytic metabolism, we have previously demonstrated that pancreatic CSCs are fundamentally oxidative, despite their increased mitochondrial fission activity [9]. In fact, there are countless examples of glycolytic cells showing fragmented mitochondria, a phenomenon which is regulated by KRAS-dependent DRP1 activation in the case of PDAC tumors [17,18]. Moreover, it has been shown that overexpression of an activating DRP1 mutant enhanced glucose uptake and lactate release in leukemia cells [30]. Interestingly, this discrepancy between the mitochondrial architecture and the cellular metabolic phenotype can also be observed in the context of stemness in embryonic and neuronal SCs, which show fragmented and fused mitochondria, respectively, linked to glycolytic metabolism [26,31].

This apparent controversy might be solved considering that mitochondrial fusion or fission may not represent merely static metabolic phenotypes, but rather dynamic processes that modulate energy expenditure in response to metabolic demands. This suggests that environmental factors and culture conditions have a strong impact on mitochondrial architecture. For instance, pancreatic beta-cells exposed to nutrient excess or physiologic uncouplers of mitochondria show increased respiration combined with fragmentation of their mitochondrial networks [32]. In these conditions, cells enhance nutrient oxidation at maximal respiratory rate while mitochondrial fission favors uncoupled respiration (decreased ATP synthesis efficiency) in order to avoid ROS overproduction and prevent oxidative damage [33]. Since we have described that sustaining low mitochondrial ROS content is essential for maintaining self-renewal and full functionality of pancreatic CSCs [9,11], we hypothesize that a similar mechanism might be operative in oxidative pancreatic CSCs to prevent excessive ROS accumulation due to elevated proton leak as compared to their differentiated counterparts (unpublished data). Despite this apparent similarity between beta-cells and CSCs, metabolism of pancreatic beta-cells is clearly distinct and adapted to the regulation of insulin secretion: on the one hand, glucose sensing is controlled by supply-driven oxidative metabolism [34]; on the other hand, beta-cells downregulate antioxidant defenses to facilitate intracellular redox signaling, which is essential for insulin secretion [35].

Consistently, the toxic effects observed with mDivi-1 in CSCs show a strong component of energetic crisis derived from the loss of mitochondrial function. Treatment with mDivi-1 in our model systems induced the accumulation of dysfunctional mitochondria with two major consequences: (1) ROS accumulation (Figure 3E), which is particularly deleterious for CSCs; and (2) diminished mitochondrial respiration, which translated into lower ATP content specifically in CSC-enriched cultures (Figure 4B–E). Importantly, DRP1 genetic loss in PDAC impaired tumor growth and showed analogous features in terms of mitochondrial dysfunction [17], validating our experimental approach using the pharmacological inhibitor mDivi-1. As we have previously reported for the inhibition of mitochondrial activity by metformin [9,10], ROS accumulation and inhibition of respiration have different consequences for PDAC cells depending on their differentiation state: proliferation blockade in bulk tumor cells, and energy crisis and cell death in CSCs, as most PDAC CSCs are unable to switch to glycolysis in order to maintain their energy levels and their increased sensitivity to oxidative stress (Appendix A) [9,11]. Importantly, other studies also described loss of mitochondrial respiration and cell death in brain and breast CSCs upon mDivi-1 treatment [28,36], suggesting that DRP1-mediated fission supports stemness in different tumor types.

One of the major consequences of DRP1 loss in PDAC is the accumulation of dysfunctional mitochondria subsequent to the blockade of mitophagy, leading to tumor growth inhibition [17]. Indeed, apart from restraining ROS production and oxidative damage by promoting uncoupled respiration, we can hypothesize that maintaining mitochondria in a fragmented state supports mitochondrial fitness by facilitating a rapid replacement of unhealthy/damaged mitochondria in CSCs. In fact, our results indicate that the accumulation of defective mitochondria is most pronounced in CD133^+^ cells (Figure 3D and Appendix A), as compared to their differentiated CD133^−^ counterparts. Most likely, this reflects their accelerated mitochondrial biogenesis [9] and mitophagy [12] rates that are required for their elevated oxidative metabolism, prone to induce oxidative damage of mitochondria. Indeed, we have demonstrated that interfering with the various stages of the mitochondrial lifecycle from biogenesis to mitophagy, including fission and fusion processes, will severely impact pancreatic CSC functionality. Reducing mitochondrial biogenesis by PGC-1α knockdown or MYC overexpression [9], interfering with mitophagy by blocking mitochondrial ISGylation via genome editing [12], or inhibiting mitochondrial fission with mDivi-1 did not only affect mitochondrial activity, but also impaired self-renewal and in vivo tumorigenicity (Figure 5). In fact, although there are still large gaps in our understanding of the interplay between mitochondrial dynamics, cell metabolism, and stemness in cancer, it is increasingly appreciated that tightly controlled mitochondrial homeostasis is essential for pancreatic CSCs functionality.

Indeed, here we provide proof-of-concept for the essential role of mitochondrial fission in PDAC stemness. Pharmacological inhibition of mitochondrial fission by mDivi-1 induced cell death in PDAC cells, with particular toxicity in the CD133^+^ subpopulation (Figure 2B–E). This decrease in the percentage of CSCs translated into diminished sphere and colony formation in vitro and tumorigenicity in vivo (Figure 5). Similar results have been described in brain and breast cancer, where mDivi-1 selectively killed the CSC subpopulation, resulting in diminished stemness and delayed tumor growth [28,36].

Importantly, inhibition of mitochondrial fission does not only decrease self-renewal and tumorigenicity, but also directly impacts the migratory and invasive abilities of PDAC cells (Figure 6A,B). Indeed, the promotion of invasiveness associated with mitochondrial fragmentation was already reported for other cancer types such as oncocytic thyroid carcinomas [37] as well as breast and lung cancers [38], where forced mitochondrial fusion inhibited colony formation, tumorigenicity and invasiveness. Indeed, invasive breast carcinomas and metastases express higher levels of DRP1 and lower levels of MFN1 compared to non-metastatic breast tumors [39]. Interestingly, and supported by our data, this effect is not transcriptional but rather functional: it has been proposed that fission maintains the required mitochondrial ATP synthesis activity to support the high energetic cost of F-actin polymerization, essential for lamellipodia formation [39].

Linking the intrinsic chemoresistance of CSCs to mitochondrial function is another emerging theme [25]. Interestingly, several previous works connect mitochondrial dynamics with chemoresistance in different cancer types [40,41,42]. In PDAC, we have shown that treatment with mDivi-1 sensitized PDAC cells to Gemcitabine treatment (Figure 6D,E). Importantly, although much more pronounced in the context of self-renewal (Figure 6E), this effect was not restricted to CSCs, since we could observe a more pronounced inhibition of proliferation by mDivi-1 and Gemcitabine treatment in the entire population of tumor cells (Figure 6D). These results suggest that the inhibition of mitochondrial fission may not only diminish the probability for disease relapse by eliminating CSCs, but may also slow down tumor progression. Consistently, mDivi-1 has also been shown to sensitize breast and ovarian cancer cells to cisplatin [41,42]. Although our data and these reports would suggest a role for mitochondrial fission in chemoresistance, studies in gynecological cancers have shown that maintaining the mitochondria in a fusion state contributes to chemoresistance, which was reversed by enforced fission [40].

Importantly, mDivi-1 treatment had no demonstratable impact on non-transformed cells (Figure 2A), in line with previous studies, and therefore suggesting a yet unchartered therapeutic opportunity for cancer [35]. While these data indicate a favorable therapeutic window for the inhibition of mitochondrial fission, for now the use of mDivi-1 should only be considered as proof-of-concept. First, the pharmacological specificity of mDivi-1 is still under debate [24,43,44] as Bordt et al. suggested that its effect on mitochondrial fission is related to mitochondrial complex I inhibition [24]. However, several findings in our study do support the specificity of mDivi-1 as mitochondrial fission inhibitor. The accumulation of dysfunctional mitochondria and enhanced ROS production (Figure 3B–F) reproduce the outcome of genetic targeting of DRP1 in PDAC [17]. In fact, accumulation of enlarged mitochondria (Figure 3A) can only be explained as a result of mitochondrial fission inhibition. Moreover, we evaluated the percentage of complex I inhibition in response to different concentrations of mDivi-1 and it mostly remained below 10% (Appendix A). Finally, while mDivi-1 has been used successfully in mice [28], its in vivo use is limited by poor lipophilicity, solubility, and pharmacodynamics [18,43]. These features would need to be improved before such a compound could be considered for further translational studies. However the lack of toxicity for non-transformed cells, its inhibitory effects on cancer cell growth, the chemosensitization, and the preferred elimination of CSCs represent convincing evidence for further advancing this concept into the clinic. Our study and previous works jointly demonstrate the suitability of mitochondrial fission inhibition as a promising strategy for developing more effective treatments for PDAC patients.

## 5. Conclusions

The process of mitochondrial fission is especially active in pancreatic CSCs, representing a novel and attractive therapeutic vulnerability for the elimination of this aggressive subpopulation of cancer cells. Pharmacological inhibition of mitochondrial fission induces accumulation of dysfunctional mitochondria, which is particularly lethal for CSCs, due to their restricted ability to activate alternative pathways for energy production. Multimodal treatments combining the inhibition of mitochondrial fission and chemotherapy could be useful to combat the still miserable survival rates of PDAC patients.

## Figures and Tables

**Figure 1 cancers-13-00698-f001:**
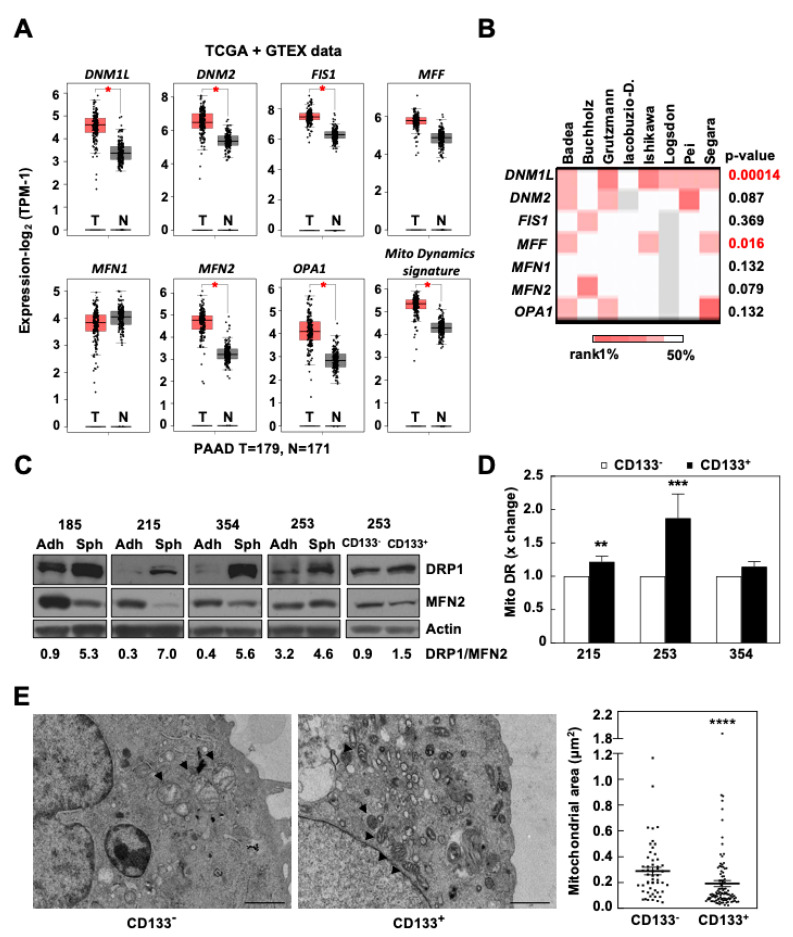
Mitochondrial fission is associated with stemness in human pancreatic ductal adenocarcinoma (PDAC). (**A**,**B**) Relative expression of genes regulating mitochondrial dynamics for normal human pancreas (N) and PDAC tissues (T), respectively, using TCGA and GTEx ((**A**) webtool GEPIA2) or other datasets ((**B**) Oncomine^TM^). TPM, transcripts per million. * *p* < 0.01. In (**B**), the color-coded rank illustrates the rank of the *p*-value for each indicated gene relative to the global list of up-regulated genes for the respective dataset (top 1%, top 10%). (**C**) Western blot for DRP1 and MFN2 in cancer stem cell (CSC)-enriched conditions (spheres (Sph) or CD133^+^ sorted cells) compared to their differentiated counterparts (cells grown in adherence (Adh) or CD133^−^ sorted cells) in four different PDAC models. The numbers below show the normalized DRP1/MFN2 expression ratio. β-actin was used as a loading control. (**D**) Mitochondrial mass as determined by flow cytometry using MitoTracker^TM^ Deep Red FM (Mito DR) in CD133^−^ and CD133^+^ cells (*n* = 6–8). Control set as 1 for fold change. (**E**) Transmission Electron Microscopy pictures (12,000×) and quantification of the mitochondrial area for CD133^−^ and CD133^+^ 185 cells, respectively (*n* = 7–17 cells or 54 vs. 107 mitochondria). ** *p* < 0.01, *** *p* < 0.001, **** *p* < 0.0001; Mann–Whitney test. Data represent averages ± S.E.M.

**Figure 2 cancers-13-00698-f002:**
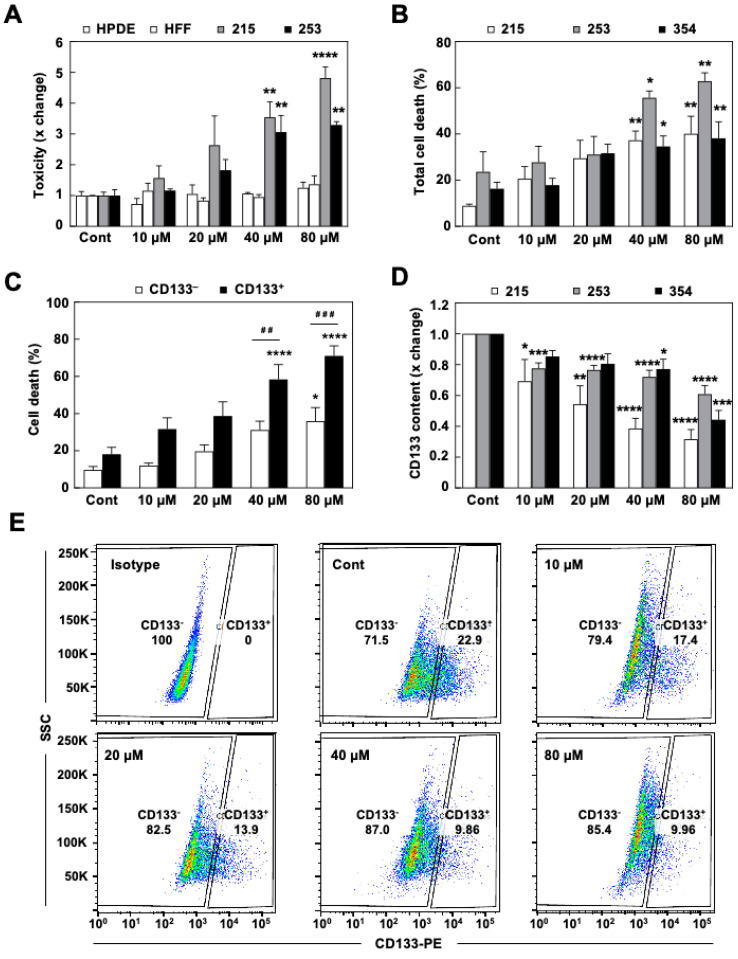
mDivi-1 treatment targets PDAC cancer cells and induces apoptosis in CSCs. (**A**) Cytotoxicity assay in response to mDivi-1 treatment for 72 h using 215 and 253 PDAC cells, non-transformed human pancreatic ductal epithelial cells (HPDE), and human foreskin fibroblasts (HFF). (**B**,**C**) Total cell death for different PDAC models after 72 h of mDivi-1 treatment, calculated for the entire cancer cell population (**B**) or for CD133^−^ and CD133^+^ cells (**C**) after staining for Annexin V-APC and Zombie Violet. (**D**) CD133 content in response to mDivi-1 treatment for 72 h, as evaluated by flow cytometric analysis in three different PDAC models. (**E**) Representative experiment in 253 cells from (**D**) * *p* < 0.05, ** *p* < 0.01, *** *p* < 0.001, **** *p* < 0.0001 vs. Control; ^##^
*p* < 0.01, ^###^
*p* < 0.001 vs. CD133^−^ cells; ANOVA with Bonferroni’s post-test (**A**–**C**) or Kruskal–Wallis with Dunn’s post-test (**D**) Data represent averages ± S.E.M. Control is set as 1 for fold change.

**Figure 3 cancers-13-00698-f003:**
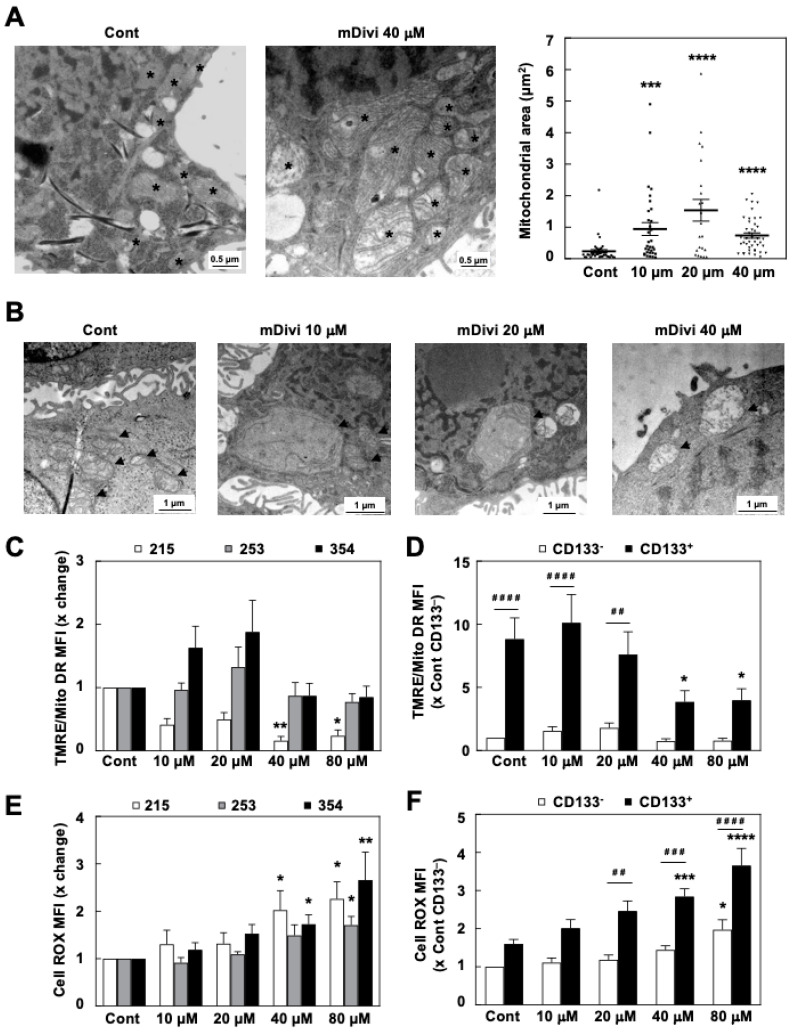
mDivi-1 treatment induces the accumulation of dysfunctional mitochondria. (**A**) TEM pictures (40,000×, left) and quantification of the mitochondrial area (right) for 354 cells with or without treatment with 40 µM mDivi-1 for 48 h (*n* = 33–43 mitochondria). *** *p* < 0.001, **** *p* < 0.0001; Kruskal–Wallis with Dunn’s post-test. (**B**) Selected areas of mDivi-1 treated cells as shown in (**A**), demonstrating altered mitochondrial morphology. (**C**,**D**) Ratio of mitochondrial activity as assessed by TMRE on mitochondrial mass (Mito DR) for bulk cancer cells (**C**) or for CD133^–^ vs. CD133^+^ cells. Pooled data for 215, 253, and 354 PDAC cells (**D**) (*n* = 4–9). (**E**,**F**) Reactive Oxygen Species (ROS) production as assessed by DCFDA for the bulk population of the indicated PDAC cells (**E**) or for CD133^−^ vs. CD133^+^ cells. Pooled data for 215, 253, and 354 cells (**F**) (*n* = 4–7). * vs. control, * *p* < 0.05, ** *p* < 0.01, *** *p* < 0.001, **** *p* < 0.0001. ^#^ vs. CD133^−^, ^##^
*p* < 0.01, ^###^
*p* < 0.001, ^####^
*p* < 0.0001. Kruskal–Wallis with Dunn’s post-test (**C**,**E**); ANOVA with Bonferroni post-test (**D**,**F**). Data shown in the figure represent averages ± S.E.M. Flow cytometry data shown represent Mean Fluorescence Intensity (MFI). Controls set as 1 for fold changes.

**Figure 4 cancers-13-00698-f004:**
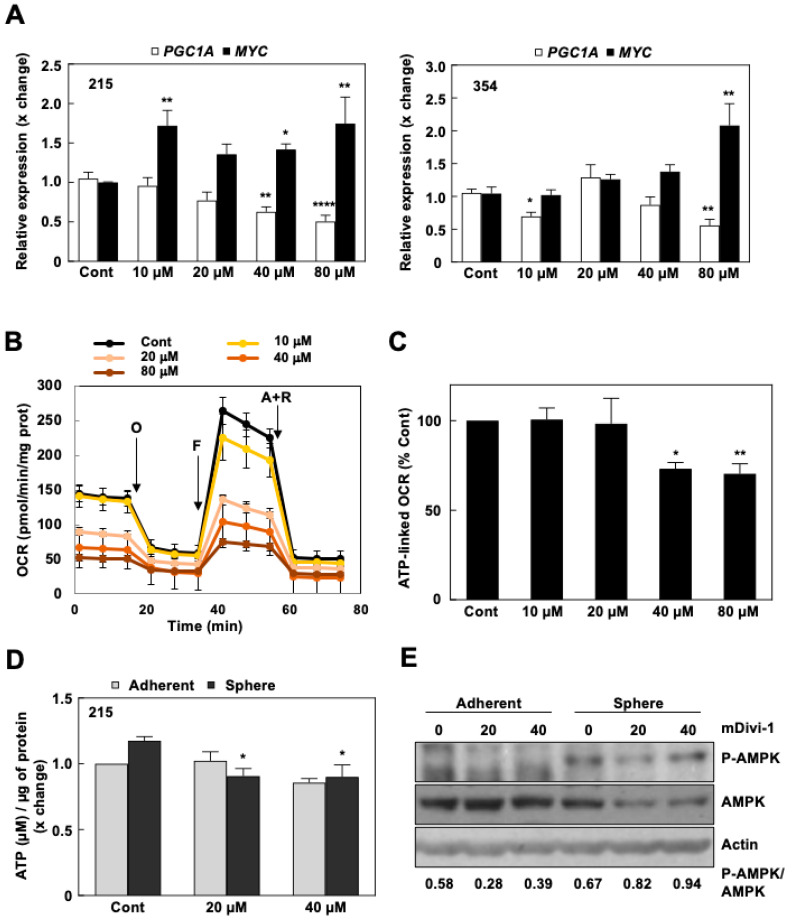
mDivi-1 treatment impairs mitochondrial respiration, provoking energy crisis in pancreatic CSCs. (**A**) Relative *PGC1A* and *MYC* gene expression in 215 and 354 cells in response to mDivi-1 treatment for 72 h (*n* = 9–16). *HPRT* expression was used as endogenous control. (**B**) Representative Mito Stress Test experiment for 253 cells treated for 48 h with mDivi-1, showing changes in the Oxygen Consumption Rate (OCR) in response to sequential treatments with Oligomycin (O), FCCP (F) and Antimycin A + Rotenone (A+R). (**C**) ATP-linked respiration measured after 48 h of mDivi-1 treatment. Pooled data for 215, 253, and 354 PDAC cells (*n* = 8–12). (**D**) Quantification of ATP content/μg protein in sphere cultures of 215 cells after 48 h of mDivi-1 treatment (*n* = 4). (**E**) Western blot for AMPK and *p*-AMPK expression in 253 cells cultured in either adherent or sphere conditions after 72 h of mDivi-1 treatment. Numbers below represent the densitometric analyses of the normalized ratio of P-AMPK on AMPK. β-actin was used as a loading control. * *p* < 0.05, ** *p* < 0.01, **** *p* < 0.0001; Kruskal–Wallis with Dunn’s post-test (**A**); ANOVA with Bonferroni post-test (**C**,**D**). Data shown in the figure represent averages ± S.E.M. Controls set as 1 or 100% for fold changes.

**Figure 5 cancers-13-00698-f005:**
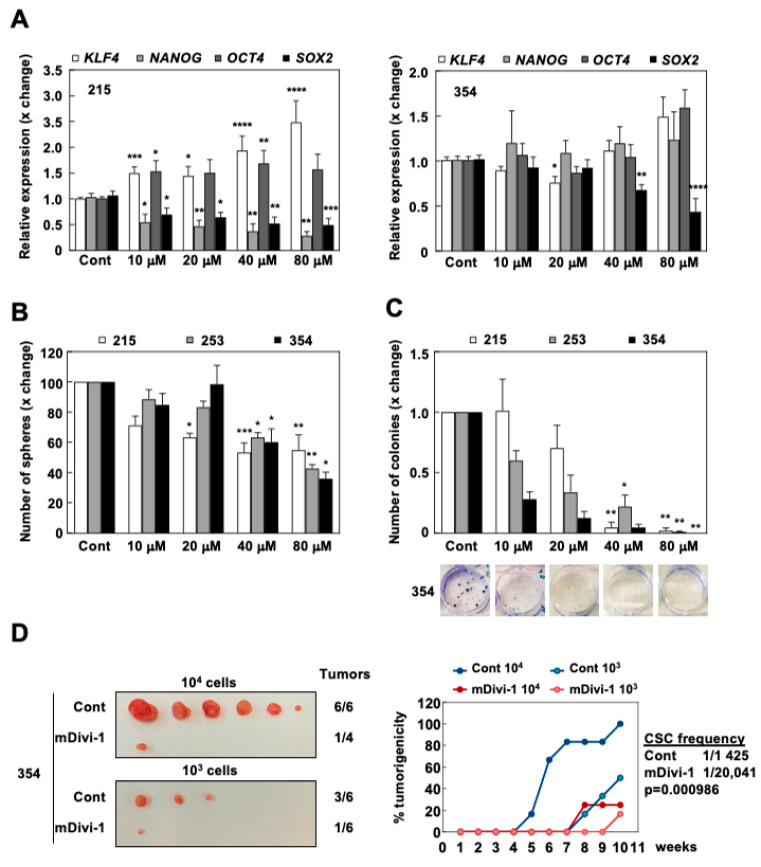
mDivi-1 treatment inhibits self-renewal and in vivo tumorigenicity. (**A**) Relative pluripotency genes expression in 215 and 354 cells treated with 40 µM mDivi-1 for 72 h (*n* = 5–7). (**B**) Number of spheres formed after seven days with or without mDivi-1 treatment (*n* = 2–6). (**C**) Colony formation during 21 days with or without mDivi-1. Upper panel, quantification. Lower panel, representative images for 253 cells (*n* = 3–6). (**D**) In vivo extreme limiting dilution assay (ELDA). Subcutaneous injection of 10^3^ or 10^4^ 354 cells pretreated for 72 h with 40 µM of mDivi-1. Left, images of the tumors obtained after 10 weeks. Right, percentage of tumorigenicity over time for each group, allowing for CSC frequency (1 CSC/x total cancer cells) estimation (*n* = 6 tumors per group). Data shown in the figure represent averages ± S.E.M. * *p* < 0.05, ** *p* < 0.01, *** *p* < 0.001, **** *p* < 0.0001; Kruskal–Wallis with Dunn’s post-test. Controls set as 1 or 100% for fold changes.

**Figure 6 cancers-13-00698-f006:**
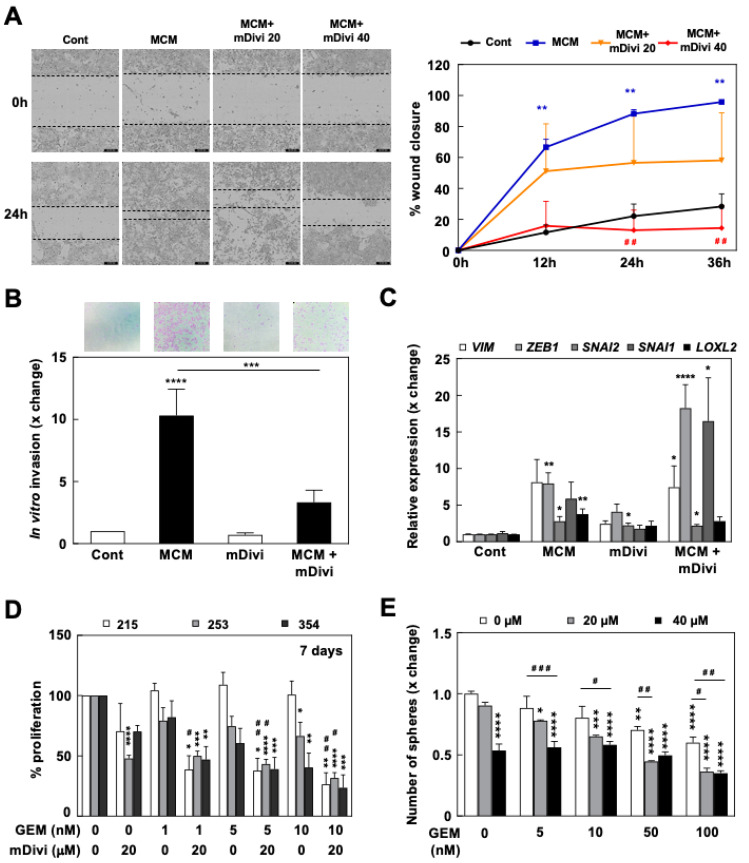
mDivi-1 treatment inhibits invasiveness and sensitizes PDAC cells to Gemcitabine. (**A**–**C**) In total, 354 cells were pretreated for 48 h with M2-polarized macrophage-conditioned medium (MCM) to induce EMT in the presence or absence of mDivi-1. (**A**) Effects of mDivi-1 treatment on migration via wound healing assay visualized in an IncuCyte^®^ at the indicated times and doses. Left, representative pictures at 0 and 24 h after scratch. Right, quantification (*n* = 3). (**B**) Invasion assay in Boyden’s chamber of MCM-treated 354 cells after treatment with mDivi-1 40 µM. (*n* = 6). Top, representative images of crystal violet stained cells after 24 h of migration. Bottom, quantification. (**C**) Relative EMT gene expression (pooled data of 215 and 354 cells, *n* = 4). (**D**) Evaluation of the proliferation rate of the indicated PDX cultures after seven days of treatment with Gemcitabine (nM) with or without 40 μM mDivi-1 (*n* = 3–7). (**E**) Sphere formation ability in the presence of Gemcitabine (nM) and/or mDivi-1 in 185 cells (*n* = 4). * vs. control (DMSO treatment) condition, * *p* < 0.05, ** *p* < 0.01, *** *p* < 0.001, **** *p* < 0.0001; # vs. MCM or Gemcitabine treatment alone, ^#^
*p* < 0.05, ^##^
*p* < 0.01, ^###^
*p* < 0.001. ANOVA with Bonferroni post-test. Data shown in the figure represent averages ± S.E.M. Controls set as 1 or 100% for fold changes.

**Table 1 cancers-13-00698-t001:** List of primers used for real time PCR.

Gene	Forward Primer	Reverse Primer
*HPRT*	TGACCTTGATTTATTTTGCATACC	CGAGCAAGACGTTCAGTCCT
*C-MYC*	CCCGCTTCTCTGAAAGGCTCTC	CTCTGCTGCTGCTGCTGGTAG
*PGC-1A*	TGACTGGCGTCATTCAGGAG	CCAGAGCAGCACACTCGAT
*NANOG*	AGAACTCTCCAACATCCTGAACCT	TGCCACCTCTTAGATTTCATTCTCT
*OCT3/4*	CTTGCTGCAGAAGTGGGTGGAGGAA	CTGCAGTGTGGGTTTCGGGCA
*SOX2*	AGAACCCCAAGATGCACAAC	CGGGGCCGGTATTTATAATC
*KLF4*	ACCCACACAGGTGAGAAACC	ATGTGTAAGGCGAGGTGGTC
*LOXL2*	GGCACCGTGTTGCGATGACGA	GCTGCAAGGGTCGCCTCGTT
*SNAIL*	GCTCCTTCGTCCTTCTCCTC	TGACATCTGAGTGGGTCTGG
*SLUG*	GTGTTTGCAAGATCTGCGGC	TTCTCCCCCGTGTGAGTTCT
*VIM*	GACAATGCGTCTCTGGCACGTCTT	TCCTCCGCCTCCTGCAGGTTCTT
*ZEB1*	GATGATGAATGCGAGTCAGATGC	CTGGTCCTCTTCAGGTGCC

## Data Availability

Not applicable.

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
