# Peer review of "Inhibition of Mitochondrial Dynamics Preferentially Targets Pancreatic Cancer Cells with Enhanced Tumorigenic and Invasive Potential"

_cancers, 2021, doi:10.3390/cancers13040698_

Round 1
Reviewer 1 Report
I do not object against publication.
Author Response
We thank reviewer 1 for supporting our work
Reviewer 2 Report
Courtois and de Luxán-Delgado provided an impressive body of evidence for the utility of the mitochondrial fission protein Drp1 inhibitor mDivi-1 as an elegant therapeutic strategy against pancreatic ductal adenocarcinoma (PDAC). Stemming from previous experiments revealing that mitochondrial fragmentation may represent an Achilles' heel in various cancer types, the authors have now intended to shed more light on the feasibility of targeting Drp1 with mDivi-1 in the context of PDAC. The involvement of mitochondrial dynamics was initially hinted at by bioinformatics approaches and in-depth data mining. The authors went on to establish the key paradigm that PDAC cancer stem cell (CSC) populations derived from patient-derived xenograft (PDX) primary in vitro cell cultures had upregulated levels of Drp1 over to those observed for the mitochondrial fusion protein Mfn2. This molecular basis was causally linked to a severe mitochondrial fragmentation phenotype evidenced by the use of transmission electron microscopy. Importantly, the bioenergetic configuration of PDAC CSCs rendered these cells profoundly sensitive to micromolar concentrations of mDivi-1. Upon mDivi-1 treatment, PDAC CSCs displayed markers of severe mitochondrial dysfunction with PGC-1α and c-MYC playing the prominent role of metabolic regulators. Indeed, further analyses provided support for the promising potential of mDivi-1 in targeting this characteristic vulnerability of PDAC CSCs. As a proof-of-principle, mDivi-1 administration was shown to interfere with the self-renewal capacity and the oncogenic potential of PDAC CSCs in an in vivo PDX mouse tumor model. Critically, the authors have incorporated into their study important control to exclude off-target effects of mDivi-1 on electron transport chain function. In addition to experimental excellence, the manuscript features with highly advanced and mature writing style and superb composition. As such, this brilliant work therefore will likely gain an extraordinary scientific recognition worldwide. To sum up, gaining mechanistic insights into the role of mitochondrial shape changes in specific niches of cancer cells will allow researchers as well as clinicians to advance further the introduction of mDivi-1 into the clinics as a prospective therapeutic avenue.
1) Please either replace "gemcitabine with "Gemcitabine" (lines 100, 324, 325, 329, 433, 436) or "Gemcitabine with "gemcitabine" (lines 44, 308, 316, 317, 482) for increased consistency.
2) Please remove bold formatting in "Juan A. Rubiolo5, Laura Sánchez5" (line 6).
3) The resolution of Figure 1A is poor. Please replot with sufficient pixel density and/or use larger font so that:
a) individual data points are clearly seen
b) asterisk can be distinguished to allow gauging significance
c) each single word in each x-axis title becomes clearly readable
4) Please indicate more clearly that OncomineTM was used to generate Figure 1B in its respective figure legend.
5) In addition, please specify what does color coding (rank percentage) represent in Figure 1B in its respective figure legend.
6) Please define abbreviation for "DR" (Figures 1D, S1E, S3A 2x, S3B), "MFI" (Figures 3C-F), "TPM" (Figures S1A,B), and "HR" (Figure S1A,D).
7) Please change "associated to" to "associated with" (lines 116, 271, 421).
8) Please replace "Actin" (lines 122, 263) and "actin" (line 519, supplementary legend to Figure S3 2x) with "ß-actin".
9) Please declare that "Control set as 1 for fold." in the legend of Figure 1D, "Controls set as 1 for fold." in the legends of Figures 2, S3, and "Controls set as 100% for fold changes" in the legend to Figures S4B-D.
10) Please refer to "Annexin V" (line 185) and "Annexin-V" (line 553) as "Annexin V-APC".
11) Please improve resolution of and/or enlarge Figure 2E so that all data points, axis numbering, gate labels, and signal percentages are all clearly readable.
12) The logic of the sentence "Moreover, mDivi-1 treatment increased the expression of the mitochondrial respiratory chain complexes (Figure S3C), corroborating the accumulation of mitochondria that can not undergo elimination via mitophagy" (line 199) is not clear. Please explain or rephrase. In the same respect, it is contradictory that increased expression of the mitochondrial respiratory chain complexes (Figure S3C) is accompanied by the decrease in respiration (Figure 4B,C) and mitochondrial dysfunction (Figure 3B) after mDivi-1 treatment. Please address this dichotomy in the Discussion section.
13) Please annotate Figure 3B and provide at least one control mitochondrion image of similar scale to mDivi-1-treated for better comparison.
14) Please provide confidence interval definition for ##, ###, and ### in the legend to Figure 3 and #, ##, and ### in the legend to Figure 6.
15) A care has to be taken in distinguishing TMRE fluorescence and mitochondrial membrane potential as these are two different measures. Please reserve the term "mitochondrial membrane potential" only for those experiments, in which this quantity is expressed in millivolts (lines 217, 221).
16) The statement "Expectedly, the observed drop in ATP-linked respiration translated into a significant drop in ATP content, which again was mostly confined to CSC-enriched cultures" (line 242) is not accurate as the decrease in ATP content did not reach significance for adherent cells (Figure 4D). Please revise.
17) Although the authors claim that "Notably, these cells were not capable of efficiently increasing glycolysis in order to compensate for the loss of ATP production upon mitochondrial inhibition" (line 244), the glycolytic reserve of 253 cells significantly increased by more than 50% following 80 uM mDivi-1 treatment (Figure S4C). Please address this discrepancy in the text.
18) Please indicate whether Figure 4A relates to protein or mRNA expression in its figure legend.
19) Please indicate concentrations of Oligomycin, FCCP, Rotenone and Antimycin used in Figure 4B,C for each cell type. Also, do the authors actually mean Antimycin A or a mixture of Antimycins?
20) The fact that the total level of AMPK decreases while its phosphorylation status remains unchanged with increasing mDivi-1 concentration thereby underlying higher phosphorylated to total AMPK ratio (Figure 4E) should be commented upon in the text.
21) Despite the authors claim that "Controls set as 1 for fold changes" (line 265), the control was set as 100% in Figure 4C. Please replace with "Controls set as 1 or 100% for fold changes."
22) Please remove "***p<0.001" from the legend of Figure 4 and add confidence interval definition for **** instead.
23) The authors claim that "mDivi-1 treatment dose-dependently reduced the number of formed spheres by >50% 277 compared to the untreated condition" (line 277), but if true,
this statement would suggest that they have chose a weak control for the purpose of this experiment since it should be the treatment with DMSO as the solvent for mDivi-1 used as a control condition (Figure 5B). Please double-check validity of this sentence or provide DMSO controls.
24) Please change "PDX215" to "215" (line 326).
25) Please comment briefly on the prospects of using combined mDivi-1 and Gemcitabine regimen in vivo akin to Figure 5D in the Discussion section.
26) Please change "correlating to" to "correlating with" (line 348).
27) Reference 33 seems to be out of the pancreatic cell context discussed in the given sentence (line 368).
28) The parallel between beta-cell and pancreatic CSC antioxdiative mechanism through uncoupling that the authors put forward in the Discussion section (lines 371-380, 394-395) is not valid for the following reasons:
a) Whereas it holds true that ROS accumulation increased (Figure 3F) and mitochondrial output decreased (Figures 4B-D), mitochondrial membrane potential actually decreased (Figures 3D) following Drp1 inhibition of CD133+ cells arguing against uncoupling under the basal state.
b) In contrast to CSCs as well as other cell types, beta-cells utilize a supply-driven rather than a demand-driven control of their oxidative metabolism to sustain their glucose-sensing function for insulin secretion (Nature 414:807).
c) Another feature specific for insulin-secreting beta-cells is the downregulation of antioxidant defense components to allow for redox signaling critical during insulin secretion, which may yet constitute another difference between CSC- and beta-cell-specific metabolic programs (Antioxid Redox Signal 14:489).
Please provide further evidence and/or add some of these counterarguments into the discussion.
29) Please replace "(Figure S5)" with "(Figure S4)" (line 388).
30) Please change "age promoting" to "age by promoting" (line 395).
31) Please add technical details on sorting of CD133+ cells into the Materials and Methods section.
32) Also, please provide information on the Boyden’s chamber used for the invasion assay in Figure 6B in the Materials and Methods section
33) Please indicate solvent used for Gemcitabine, Oligomycin, FCCP, Rotenone and Antimycin in the Methods section.
34) Please reveal for how many maximum passages were PDX cells cultured for experiments?
35) Please specify catalog numbers for RPMI (line 469), DMEM/F12 (line 471), DMEM (line 476), Pierce BCA Protein Assay Kit (line 576), non-adherent plates (line 579), M2-polarized macrophages (MCM) (line 590), and PET membrane invasion chambers (line 592).
36) Please provide reference/software/webserver for the Cox Proportional Hazards model used to derive disease-free survival curves (line 491).
37) Please include catalog numbers for the secondary Western blot antibodies used in the 4.5. Immunoblots section.
38) Please state how many microliters or what percentage of DMSO was used as a control for each assay in 4.2. Treatments section.
39) Please indicate whether HPRT was used as the housekeeping gene internal control in 4.6 RNA and RTqPCR section?
40) Please specify whether the treatment was with mDivi-1 in "10,000 cells were seeded in triplicates in different 96-well plates and treated 24h later in 200μL of supplemented DMEM/F12" (line 536)?
41) Please replace "40583100" with "405831000" and indicate Fisher Scientific as the only supplier for crystal violet (line 538).
42) Please change "eBiosciences" to "BD Biosciences" (line 554).
43) Please replace "Stress Kit" with "Stress Test Kit" (line 563).
44) Please replace "Glyco Stress kit" with "Glycolysis Stress Test Kit" (line 566).
45) Please specify the type of assay used for estimating protein content in the 4.10. XF Extracellular Flux Analyzer Experiments section (line 571).
46) What do the authors mean by ultrapure water (line 573)? Please provide its catalog number or indicate its conductance in MΩ.
47) Please change "Crystal violet" to "crystal violet" (line 594).
48) Please indicate the method of evaluating tumor size in the 4.16. In vivo Extreme Limiting Dilution Assay (ELDA) section.
49) Please replace "SEM" with "S.E.M." (line 617).
50) Please change "L.P.P" to "L.P-P" (lines 633, 634).
51) Please fill out "Institutional Review Board Statement" (line 648), "Informed Consent Statement" (line 649), and "Data Availability Statement" (line 650) or leave as "Not applicable."
52) Please indicate the respective institution in which José Luis Aldea and Pilar Espiau work (line 652).
53) Please indicate what do the dotted red and blue lines mean in Figures S1A,D?
54) Please provide full p-values for Figures S1B,C.
55) Please remove italics formatting in "EMT signature" in the y-axis title of Figure S1C. Also, would it please be possible to replot this figure with all data points having identical size (thickness) as that used in Figure S1B?
56) Please normalize data to 1 instead of 0.5 in Figure S1E and declare that "Control set as 1 for fold." in the respective figure legend.
57) Please provide information on O2 level and the experimental set up used for hypoxic conditioning in Figure S2C and in the Materials and Methods section. Please state whether oxygen sensor was used to monitor O2 level.
58) Please relabel y-axis of Figure S3B so that its marks can be unambiguously discerned.
59) Please provide error bars to C IV signal in Figure S3C.
60) Please remove ", ***p < 0.001, ****p < 0.0001" from the legend to Figure S3 and add confidence interval definition for # (Figure S3B).
61) Please add control data points (Rotenone only) and indicate whether basal or ATP-linked respiration was assayed, what concentration of Rotenone was used, and how long after Rotenone addition was mDivi-1 added in Figure S4A. Perhaps a representative trace would help understand how this measurement was performed.
62) Please change "of the glyco stress kit" to "using the XF Glycolysis Stress Test Kit" in the supplementary legend to Figures S4B-D.
63) Please rename the supplementary file as "Courtois & de Luxán-Delgado et al suppl figs 110121".
Author Response
Please, see attachment

Reviewer 3 Report
Courtois et al. have compiled a very nice manuscript showing in a proof-of-concept that targeting mitochondrial dynamics in pancreatic cancer specifically induces energy crisis and subsequently cell death in cancer stem cells. Treatment with an inhibitor of mitochondrial fission (mDivi) reduces pro-metastatic characteristics and tumorigenicity in PDX models. Additionally, the authors combined gemcitabine with the inhibitor mDivi and could show that mitochondrial targeting sensitises three different PDx model to gemcitabine treatment.
As I very much enjoyed reading the manuscript, I only have a few issues that should be taken care of.
- Figure S2: This figure needs to be redone; displayed should be the EC50 (half maximal effective concentration) not IC50 (half maximal inhibitory concentration). For calculating the EC50, the curve should be plotted as the response (viability) versus the logarithm of the concentration. Please recalculate the EC50 values and remember the x-axes titles.
- Figure 1B: Please add to the legend an explanation about the rank.
- Figure S3C: The authors claim in the text “mDivi-1 treatment increased the expression of the mitochondrial respiratory chain complexes”. However, due to the error bars in the figure, this does not look convincing to me. The blots don’t convince me either. Please rephrase the result.
- Figure 6 A-B: Please add which of the PDX model was used for these figures. Did all PDX show the same result?
Author Response
Please, see attachment
